# Synthesis and Improved Photoluminescence of SnF_2_-Derived CsSnCl_3_-SnF_2_:Mn^2+^ Perovskites via Rapid Thermal Treatment

**DOI:** 10.3390/ma16114027

**Published:** 2023-05-28

**Authors:** Jisheng Xu, Haixia Wu, Xinye Lu, Yaqian Huang, Jianni Chen, Wendi Zhou, Zewen Lin, Jie Song, Hongliang Li, Rui Huang

**Affiliations:** School of Materials Science and Engineering, Hanshan Normal University, Chaozhou 521041, China; jishengxu2023@126.com (J.X.); 20220063@hstc.edu.cn (H.W.); lu15768524048@126.com (X.L.); h15976709657@126.com (Y.H.); a1963638106@126.com (J.C.); a54495597@126.com (W.Z.); zewenlin@126.com (Z.L.); songjie@hstc.edu.cn (J.S.)

**Keywords:** CsSnCl_3_, Mn^2+^, photoluminescence, rapid thermal treatment, SnF_2_-derived

## Abstract

We report a rapid synthesis method for producing CsSnCl_3_:Mn^2+^ perovskites, derived from SnF_2_, and investigate the effects of rapid thermal treatment on their photoluminescence properties. Our study shows that the initial CsSnCl_3_:Mn^2+^ samples exhibit a double luminescence peak structure with PL peaks at approximately 450 nm and 640 nm, respectively. These peaks originate from defect-related luminescent centers and the 4T^1^→6A^1^ transition of Mn^2+^. However, as a result of rapid thermal treatment, the blue emission is significantly reduced and the red emission intensity is increased nearly twofold compared to the pristine sample. Furthermore, the Mn^2+^-doped samples demonstrate excellent thermal stability after the rapid thermal treatment. We suggest that this improvement in photoluminescence results from enhanced excited-state density, energy transfer between defects and the Mn^2+^ state, as well as the reduction of nonradiative recombination centers. Our findings provide valuable insights into the luminescence dynamics of Mn^2+^-doped CsSnCl_3_ and open up new possibilities for controlling and optimizing the emission of rare-earth-doped CsSnCl_3_.

## 1. Introduction

In recent years, all-inorganic lead halide perovskite (CsPbX_3_, X=Cl, Br, I) has garnered significant attention in optoelectronic devices due to its outstanding photoelectric properties, such as high photoluminescent quantum yield (PLQY), excellent defect tolerance, and tunable light emission [1,2,3,4]. Currently, LED devices based on CsPbX_3_ have surpassed 20% external quantum efficiency. Further, the stability issues related to lead-based perovskites have been effectively improved via packaging, increasing their lifespan from a few hours to hundreds of hours. However, the high toxicity of lead poses a threat to human health and the ecological environment, thereby impeding their practical use. Therefore, there is a rising demand for non-toxic or less toxic elements to replace lead, such as equivalent substitution (Sn^2+^, Ge^2+^) and isovalent substitution (Bi^3+^, Ag^+^) to form lead-free perovskite luminescent materials [5,6,7,8,9,10]. It is proposed to replace Pb in lead–halide perovskite with divalent Sn and Ge cations as they both satisfy the prerequisites for coordination and charge balance. Sn^2+^ has an ion radius comparable to Pb^2+^ (1.35 Å for Sn^2+^ and 1.49 Å for Pb^2+^), resulting in the avoidance of significant lattice vibration caused by substitution. Additionally, Sn^2+^ possesses a similar ns^2^ electronic configuration to Pb^2+^, making it possible to form a perovskite with the formula ASnX_3_, akin to lead-based counterparts [11,12,13,14,15,16,17]. However, the formation energy of defects in Sn-based perovskites is relatively low (~250 meV), making it easy to form a defect density of up to 10^19^ cm^–3^ in Sn-based perovskites, which results in a PLQY of as low as 0.14% in these Sn-based perovskites. Additionally, Sn^2+^ is susceptible to oxidation into Sn^4+^ in air, leading to structural instability of perovskites and luminescence quenching. Recently, Zhang et al. employed SnF_2_ instead of easily oxidized SnBr_2_ as a tin source and effectively improved the structural stability of Cs_4_SnBr_6_ perovskite through introducing F to inhibit the oxidation of Sn^2+^ in tin-based perovskite [18]. Despite the progress, the luminescence stability of cesium tin chloride perovskite is still poor, and the luminescence efficiency is too low to meet the application requirements. In particular, the synthesis and photoluminescence properties of SnF_2_-derived cesium–tin perovskite (CsSnCl_3_-SnF_2_) have not been reported so far.

On the other hand, impurity doping is considered an effective method to improve the luminescence properties and stability of tin-based perovskites. For instance, Dawson et al. successfully improved the stability of tin-based perovskites through replacing Sn^2+^ sites with Mn^2+^ [19]. Similarly, Hou et al. discovered that Mn^2+^ doping could improve the electron energy level structure and stability of tin halide perovskites and also greatly enhance the luminescence efficiency of CsSnCl_3_:Mn^2+^ to around 2% [20]. However, the low-temperature chemical synthesis of Mn-doped tin-based perovskites complicates the full activation of the Mn^2+^ luminescent centers. Due to tin-based halide perovskite materials’ poor thermal stability, conventional thermal annealing may not effectively activate the Mn ion luminescent centers in this kind of material. Rapid thermal treatment (RTT), a transient process that is characterized by rapid heating and cooling, provides an alternative way to thermally activate the Mn ion luminescent centers of tin-based perovskites. Here, we describe a rapid synthesis method for SnF_2_-derived CsSnCl_3_:Mn^2+^ and investigate the effect of RTT on the photoluminescent (PL) properties of CsSnCl_3_:Mn^2+^ perovskites. Pristine CsSnCl_3_:Mn^2+^ displays a double luminescence peak structure, which includes PL peaks at around 450 nm and 640 nm, respectively. These peaks originate from defect-related luminescent centers and the 4T^1^→6A^1^ transition of Mn^2+^, respectively. We found that RTT significantly reduced the blue emission and increased the red emission intensity nearly twofold compared to the pristine sample. Furthermore, the Mn^2+^-doped samples exhibited excellent thermal stability after RTT processing. We discuss the improved PL based on our analysis of PL excitation spectra, XRD patterns, and PL decay traces.

## 2. Materials and Methods

Mn^2+^-doped CsSnCl_3_ were synthesized using a water-assisted wet ball-milling method. The reactant precursors included cesium chloride (4 mmol, CsCl, Aladdin, 99.9%), stannous fluoride (1 mmol, SnF_2_, Macklin, 99.9%), ammonium bromide (1 mmol, NH_4_Cl, Aladdin, 99.99%), and manganous chloride (MnCl_2_, Macklin, 99%). To obtain Mn^2+^-doped CsSnCl_3_ with different levels, the molar ratios of CsCl, SnF_2_, and NH_4_Cl were maintained at 1:1:2 mmol, whereas the molar ratio of MnCl_2_ was kept at 1 mmol. Firstly, the precursors were loaded into a jar and mixed with 60 μL of deionized water. A ball milling process was then performed for 30 min at a speed of 600 rpm. The resulting product was subsequently dried in a vacuum-drying oven for 120 min at room temperature and annealed at 200 °C using a simple rapid thermal treatment (RTT) process. The RTT process was carried out on a rapid thermal processor, heating the sample to the annealing temperature at a rate of 10 °C s^−1^. After remaining at the annealing temperature of 200 °C for 60–300 s, the system was rapidly cooled down to room temperature. Upon cooling to room temperature, the Mn^2+^-doped CsSnCl_3_ powder was obtained via ball milling for 30 min at a speed of 600 rpm. Figure 1 illustrates the Mn^2+^-doped CsSnCl_3_ synthesis process using water-assisted ball-milling at room temperature, followed by the RTT process. The crystal structures of Mn^2+^-doped CsSnCl_3_ were characterized using X-ray diffraction (XRD) (Bruker D8 Advance, Karlsruhe, Germany) at 35 kV and 35 mA. The compositions of the Mn^2+^-doped CsSnCl_3_ were analyzed through energy dispersive spectroscopy (EDS) (Bruker EDS QUANTAX, Karlsruhe, Germany). The structure of the Mn^2+^-doped CsSnCl_3_ sample (S-Mn^2+^-doping) was characterized via scanning electron microscopy (SEM, Hitachi SU5000, Tokyo, Japan). PL measurements, including temperature-dependent PL spectra, PL excitation (PLE) spectra, and time-resolved PL spectra, were carried out using a PL spectrometer (FLS1000, Edinburgh Instrument, Livingstone, UK).

## 3. Results and Discussion

Figure 2a displays the SEM image obtained from the Mn^2+^-doped CsSnCl_3_ powders prior to the RTT process. The EDS elemental mapping of the Mn^2+^-doped CsSnCl_3_ reveals a well-distributed presence of Cs, Pb, Cl, Mn, and F elements. Figure 2b indicates that the CsSnCl_3_ powder retains a uniform distribution of Cs, Sn, Cl, Mn, and F elements even after the RTT process.

Figure 3a depicts the PL spectrum obtained from the Mn^2+^-doped CsSnCl_3_ powders prior to the RTT process. Pristine CsSnCl_3_:Mn^2+^ shows a double luminescence peak structure with PL peaks at ~450 nm and ~640 nm, respectively. As portrayed in the inset of Figure 3a, the 580 nm PL band from CsSnCl_3_ powders without Mn^2+^ doping is due to the radiative recombination of self-localized excitons [21]. The addition of Mn^2+^ doping into CsSnCl_3_ results in a significant 60 nm redshift in the PL, which is ascribed from the 4T^1^→6A^1^ transition of Mn^2+^ [22]. To comprehend the origin of the blue emission, the temperature-dependent PL spectra of the pristine CsSnCl_3_:Mn^2+^ were measured from 78 to 298 K. As illustrated in Figure 3b, the blue emission displayed an increase in PL intensity and a reduction in the full width at half maximum (FWHM) with decreasing temperature, which is due to thermal quenching. Nevertheless, its peak barely changes with temperature, suggesting a typical defect-related luminescence behavior and implying that the blue emission arises from the defect-related luminescence centers. In contrast, the red emission intensity displays almost no recognizable changes with temperature, and only the PL peak redshifts slightly to 670 nm with decreasing temperature, indicating a competitive relationship between the blue and red emission in the pristine CsSnCl_3_:Mn^2+^. To clarify the PL characteristics, PL decay curves were measured under an excitation wavelength of 375 nm (375 nm, 70 ps excitation pulses LASER), portrayed in Figure 3c,d. The PL decay curve for blue emission was well-fitted with a biexponential decay function, while the PL decay curve for red emission was well-fitted with a triexponential decay function [23]:(1)$$I\left(t\right)=I_{0}+A_{1}exp\left(\frac{-t}{\tau_{1}}\right)+A_{2}exp\left(\frac{-t}{\tau_{2}}\right)+A_{3}exp\left(\frac{-t}{\tau_{3}}\right)$$
where I_0_ represents the background level; τ_1_, τ_2,_ and τ_3_ represent the lifetime of each exponential decay component, and A_1_, A_2_, and A_3_ denote the corresponding amplitudes, respectively. Therefore, the intensity-weighted average PL lifetimes are determined using $\left(A_{1}*\tau_{1}^{2}+A_{2}*\tau_{2}^{2}+A_{3}*\tau_{3}^{2}\right)/\left(A_{1}*\tau_{1}+A_{2}*\tau_{2}+A_{3}*\tau_{3}\right)$ [24]. As demonstrated in Figure 3c, the blue emission exhibits a fast decay dynamic with a lifetime of 4.93 ns, while the red emission displays a slow decay behavior with a lifetime of 0.16 ms, which is five orders of magnitude longer than that of the blue emission. These findings adequately explain the competitive relationship between the defect-related luminescence centers and the Mn^2+^ state in the pristine CsSnCl_3_:Mn^2+^. Upon light excitation, the excitons lifted from the ground state to the excited state relax rapidly to the defect-related luminescence centers due to the fast decay dynamic, thereby significantly intensifying the blue emission.

The PL and PLE spectra of CsSnCl_3_:Mn^2+^ powders with and without the RTT process are depicted in Figure 4. Interestingly, it was observed that the RTT process led to a remarkable enhancement in red emission while significantly reducing the blue emission. Moreover, the intensity of the red emission increased with an increase in RTT time. Specifically, after the RTT process, the excitation peaks of CsSnCl_3_:Mn^2+^ at 355, 418, and 469 nm, corresponding to the ^6^A_1_(^6^S)→^4^T_2_(^4^D), ^4^T_2_(^4^G), and ^4^T_1_(^4^G) transitions of the Mn^2+^ ion [25], respectively, were observed, as shown in Figure 4. This strongly indicates that the RTT process effectively activates Mn^2+^ in CsSnCl_3_, leading to a significant enhancement in excited-state density, as revealed in Figure 4. Therefore, the enhancement in excited-state density is suggested to be responsible for the improved red emission. Interestingly, from Figure 4, it was found that the blue emission wavelength overlapped with the transition from ^6^A_1_(^6^S)→^4^T_2_(^4^G) and ^4^T_1_(^4^G) of the Mn^2+^ ion. This indicates that the reduction in blue emission may have resulted from the energy transfer from the defect-related luminescent state to the ^6^A_1_(^6^S)→^4^T_2_(^4^G) and ^4^T_1_(^4^G) transitions of the Mn^2+^ ion, which in turn contributes to the improvement in red PL intensity. Thus, the significant enhancement in red emission can be attributed to the enhanced excited-state density as well as energy transfer between the defect-related luminescent state and the Mn^2+^ state.

Figure 5 presents the PL decay curves of the CsSnCl_3_:Mn^2+^ powders with and without the RTT process. It was found that the average lifetime rapidly increased from 0.16 ms to 4.73 ms after the RTT process. Furthermore, with an increase in the RTT processing time from 60 s to 300 s, the average lifetime gradually increased from 4.73 ms to 6.81 ms, indicating a reduction in nonradiative recombination centers. This was confirmed via the XRD patterns of the CsSnCl_3_:Mn^2+^ powders with and without the RTT process, as shown in Figure 6. For the CsSnCl_3_:Mn^2+^ powders without the RTT process, the diffraction peaks at 22.10°, 23.95°, 30.269°, and 31.142° corresponded to (400), (020), (411), and (002) crystal planes of cubic CsSnCl_3_ (PDF# 71-2023), respectively. In addition, there were still small amounts of characteristic peaks of CsCl. However, after the RTT process, the diffraction peaks corresponding to (400), (020), (411), and (002) crystal planes of cubic CsSnCl_3_ became stronger and sharper with an increase in the RTT time from 60 s to 300 s. This strongly indicates an improved crystallinity of the CsSnCl_3_:Mn^2+^ powders after the RTT process, which coincided well with the increasing lifetime with an increase in the RTT time. Evidently, the RTT not only effectively activated the Mn^2+^ but also improved the crystallinity of the CsSnCl_3_:Mn^2+^ powders, thus reducing the nonradiative recombination centers and contributing to the enhancement in red emission. Raman spectra in Figure 7 show an array of symmetric (125–175 cm^−1^) and asymmetric (185–260 cm^−1^) stretching peaks for the SnCl_3_^−^ group, and a peak at ~275 cm^−1^, attributable to symmetric Mn–Cl stretching.

To gain a deeper understanding of the PL characteristics, the temperature-dependent PL spectra of the CsSnCl_3_:Mn^2+^ powders with 300 s of RTT processing were measured, and the results are shown in Figure 8. It was observed that with a decrease in temperature, there was a significant increase in the red PL intensity along with a slight increase in the blue PL intensity. This is in contrast to the behavior observed in the CsSnCl_3_:Mn^2+^ powders without the RTT process, as shown in Figure 3b. This increase can be attributed to the energy transfer from the defect-related luminescent state to the 6A^1^(6S)→4T^2^(4G) and 4T^1^(4G) transitions of the Mn^2+^ ion. The temperature dependence of the integrated red PL intensity I_PL_(T) can be fitted using the Arrhenius equation given below [22]:(2)$$I_{PL}\left(T\right)=\frac{I_{PL}\left(T_{0}\right)}{1+\beta \exp-E_{b}/k_{B}T}$$
where I_PL_(T_0_) is the integrated PL intensity at 10 K, β is a constant related to the density of radiative recombination centers, k_B_ is Boltzmann’s constant, and E_b_ is the exciton binding energy. Using this equation, the exciton binding energy E_b_ was empirically derived to be 31.6 meV, as shown in the inset of Figure 8. This further confirms that the red emission is from the Mn^2+^ state rather than from self-trapped excitons.

To evaluate the thermal stability of the CsSnCl_3_:Mn^2+^ powders, the temperature-dependent integrated PL intensities were monitored during thermal recycling. As shown in Figure 9, both samples experienced thermal quenching of PL as temperatures were increased from 25 to 165 °C. However, for pure CsSnCl_3_ powders, a reduction of over 90% in PL intensity was observed after the heating and cooling cycles. In contrast, CsSnCl_3_:Mn^2+^ powders with RTT processing showed a slight enhancement in PL intensity after the heating and cooling cycles. Clearly, the CsSnCl_3_:Mn^2+^ powders with RTT processing displayed superior thermal and structural stability.

## 4. Conclusions

We demonstrated a rapid synthesis method for SnF_2_-derived CsSnCl_3_:Mn^2+^ perovskite and investigated the effect of RTT on its PL properties. We observed that pristine CsSnCl_3_:Mn^2+^ exhibited a double luminescence peak structure, attributed to defect-related luminescent centers and the 4T^1^→6A^1^ transition of Mn^2+^. RTT was found to significantly reduce the blue emission and enhance the red emission intensity almost twofold compared to the pristine sample. This improvement in PL was suggested to arise from increased excited-state density, energy transfer between the defect-related state and the Mn^2+^ state, as well as a reduction of nonradiative recombination centers. Furthermore, we demonstrated that the Mn^2+^-doped samples after RTT exhibit excellent thermal stability. Our findings clearly demonstrate that RTT not only can avoid thermal decomposition of tin-based halide perovskite materials during the heating process but also improve the crystallinity of CsSnCl_3_:Mn^2+^ powders, and more importantly provides an alternative method for thermal activation of the Mn ion luminescence center of tin-based perovskite, which has implications for the future development of optoelectronic devices.

## Figures and Tables

**Figure 1 materials-16-04027-f001:**
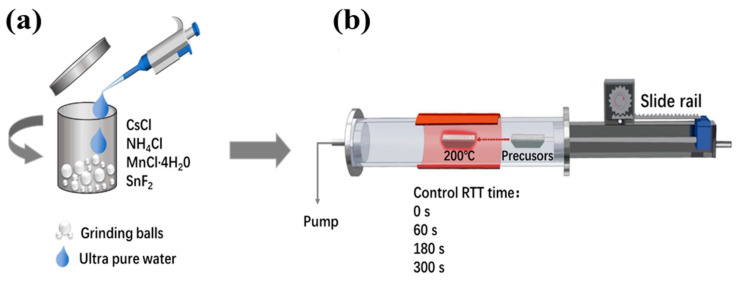
Schematic showing the method for (**a**) synthesizing Mn^2+^-doped CsSnCl_3_ and (**b**) RTT process.

**Figure 2 materials-16-04027-f002:**
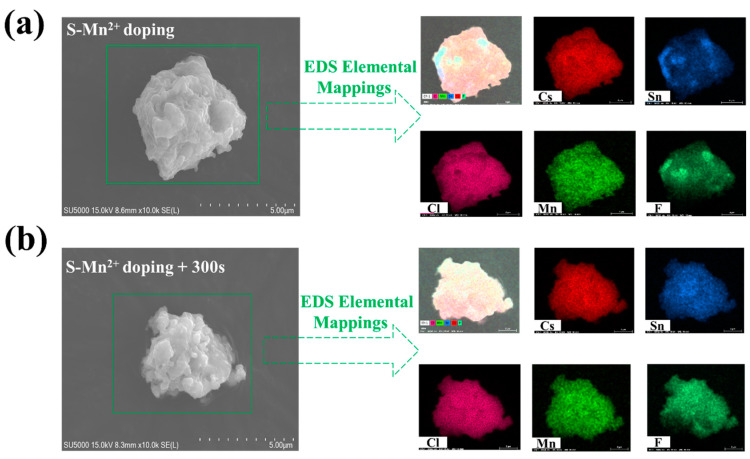
SEM image and EDS elemental maps of Cs, Sn, Mn, Cl, and F for a typical Mn^2+^-doped CsSnCl_3_ powder (**a**) before RTT process and (**b**) after RTT process for 300 s, respectively.

**Figure 3 materials-16-04027-f003:**
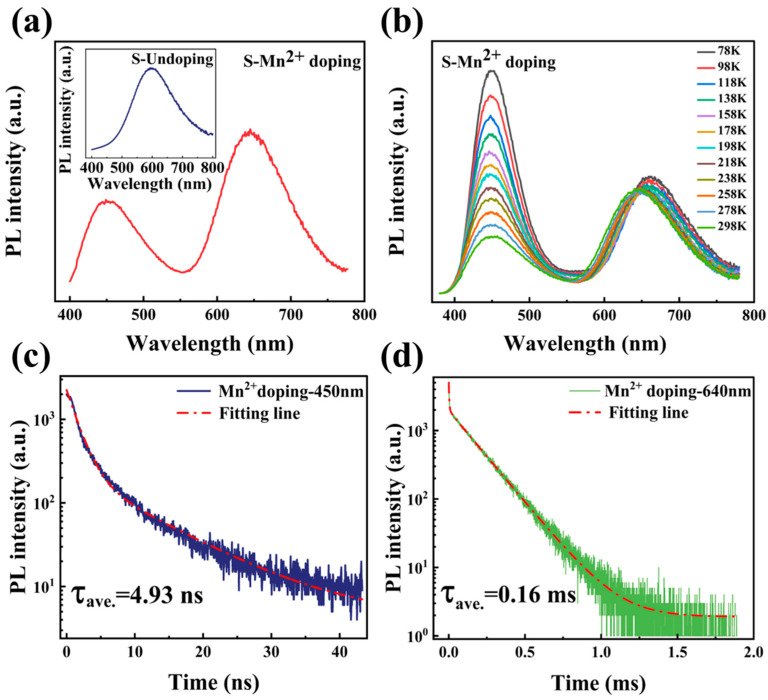
(**a**) PL spectrum from the Mn^2+^-doped CsSnCl_3_ powders without RTT process. Inset shows PL spectrum from the CsSnCl_3_ powders. (**b**) Temperature-dependent PL spectra of the pristine CsSnCl_3_:Mn^2+^ measured in the range of 78 to 298 K. Time-resolved PL decay trace of (**c**) blue PL recorded at 450 nm and (**d**) red PL recorded at 640 nm in the pristine CsSnCl_3_:Mn^2+^.

**Figure 4 materials-16-04027-f004:**
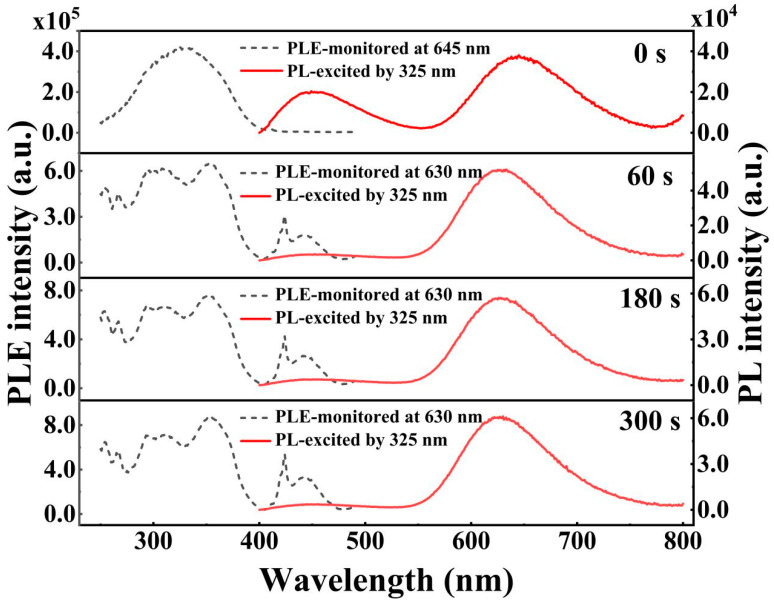
PL and PLE spectra of CsSnCl_3_:Mn^2+^ powders with and without RTT process, respectively.

**Figure 5 materials-16-04027-f005:**
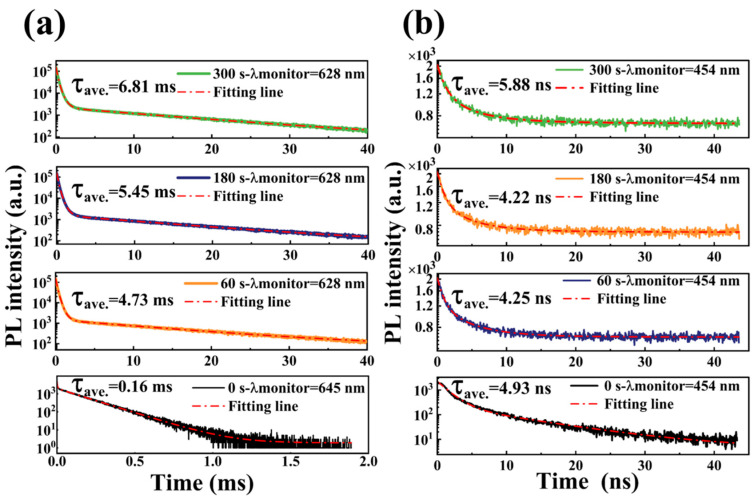
Time-resolved PL decay trace of (**a**) red PL recorded at corresponding PL peak and (**b**) blue PL recorded at 454 nm in the CsSnCl_3_:Mn^2+^ powders with and without RTT process, measured under an excitation wavelength of 375 nm (375 nm, 70 ps excitation pulses LASER).

**Figure 6 materials-16-04027-f006:**
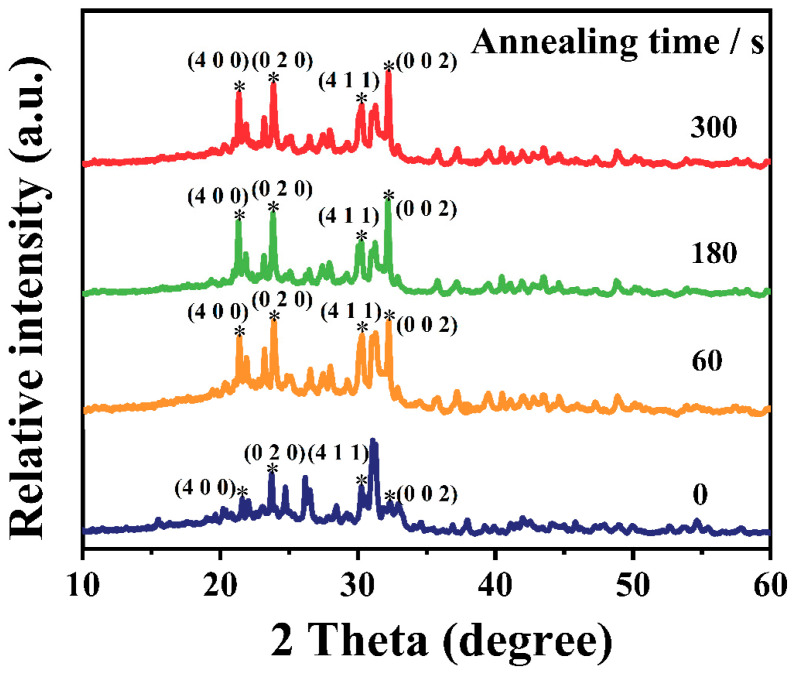
XRD patterns obtained for the CsSnCl_3_:Mn^2+^ powders with and without RTT process, The symbol “*”represents the corresponding crystal plane.

**Figure 7 materials-16-04027-f007:**
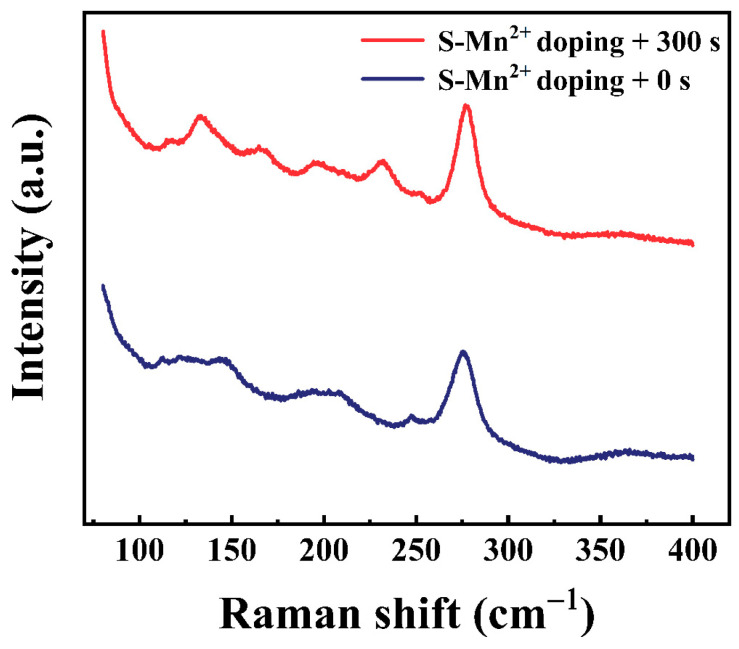
Raman spectra of CsSnCl_3_:Mn^2+^ powders with and without RTT process.

**Figure 8 materials-16-04027-f008:**
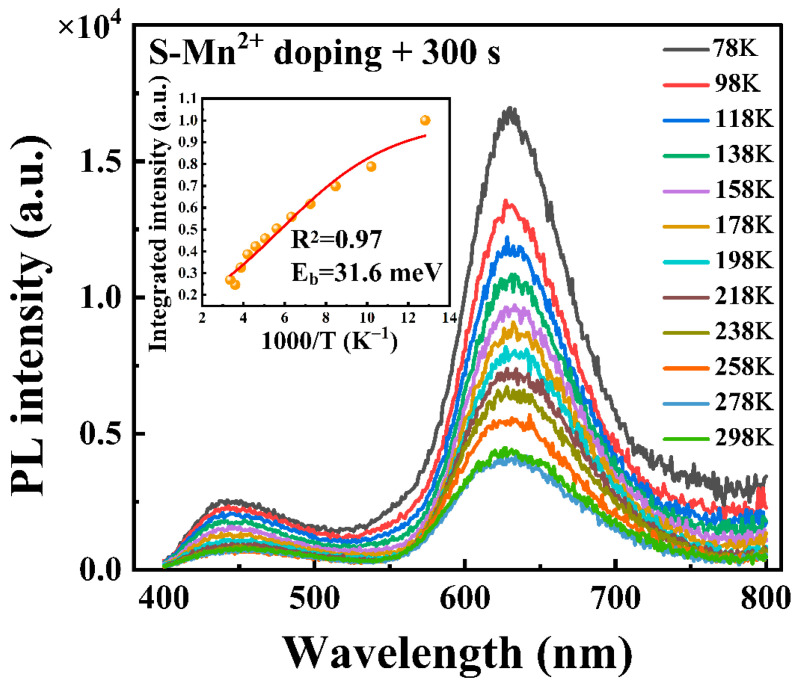
Temperature-dependent PL spectra of the CsSnCl_3_:Mn^2+^ powders with RTT processing for 300 s measured in the range of 78 to 298 K. Inset shows the integrated red PL intensities at different temperatures (orange solid symbols) and the fitting of the experimental data (red curve).

**Figure 9 materials-16-04027-f009:**
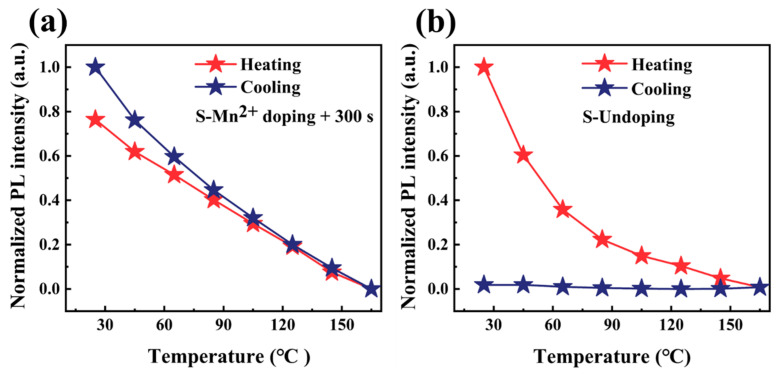
Heating and cooling cycling measurements at various temperatures: (**a**) pure **CsSnCl_3_**; (**b**) **CsSnCl_3_:Mn^2+^** powers with RTT processing for 300 s.

## Data Availability

Data underlying the results presented in this paper are not publicly available at this time but may be obtained from the authors upon reasonable request.

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
