# Peer review of "Synthesis and Improved Photoluminescence of SnF2-Derived CsSnCl3-SnF2:Mn2+ Perovskites via Rapid Thermal Treatment"

_materials, 2023, doi:10.3390/ma16114027_

Round 1

Reviewer 1 Report

This work reports the improved photoluminescence of Mn-doped CsSnCl3 perovskites synthesized by rapid thermal treatment (RTT), and SnF2 was used as a precursor of Sn. The RTT-synthesized Mn-doped CsSnCl3 exhibits both blue and red emissions, with the latter being almost two times larger compared to the pristine sample. The authors also successfully demonstrated the thermal stability and the contribution of Mn dopant in enhancing the intensity of the red emission. This work thus provides an effective pathway to engineering the PL characteristics of CsSnCl3. The manuscript is well-written. I, therefore,  recommend its publication with a minor revision. My comments are below.

Comment#1: In Figure 2 and other places, the authors keep using the term ‘S-Mn2+’, but they didn’t define it. Does ‘S’ stand for substitutional doping of Mn2+ at the Sn2+ site?

Comment#2: In the temperature-dependent PL for Mn-doped CsSnCl3 (Figure 3b), the intensity decreases with increasing temperature. I recommend authors provide at least a tentative explanation for such behavior of the PL intensity.

Comment#3: Ref. 23 is used for Equation (1). However, it is not the original paper. I suggest citing the original paper as well, which is, Lin et al. Opt. Express 21, 23416-23424 (2013)

Comment#4: Figure 4. The term ‘PLE’ was not defined. Also, in the Introduction, the acronym ‘NCs’ was used but not defined earlier.

Comment#5: Two typos in Figure 7 needed to be corrected: Ea——>Eb and Mev——> MeV.

Also, the PL intensity is observed to be increasing with decreasing temperature. However, it slightly drops at T=278 K. Can the authors provide any qualitative explanation?

Comment#6: I suggest replacing the ‘rare earth doped’ with ‘rare-earth-doped’ in the Abstract.

Author Response

1.In Figure 2 and other places, the authors keep using the term ‘S-Mn2+’, but they didn’t define it. Does ‘S’ stand for substitutional doping of Mn2+ at the Sn2+ site?

Response:

We have define the term “S-Mn2+ doping” in the revised manuscript. The “S” stand for sample. Please see Page 4, line 5.

2.In the temperature-dependent PL for Mn-doped CsSnCl3 (Figure 3b), the intensity decreases with increasing temperature. I recommend authors provide at least a tentative explanation for such behavior of the PL intensity.

Response:

We have provide a explanation for the behavior of the PL intensity with temperature. Please see Page 6, line 4-5.

3.Ref. 23 is used for Equation (1). However, it is not the original paper. I suggest citing the original paper as well, which is, Lin et al. Opt. Express 21, 23416-23424 (2013)

 Response:

We have cited the original paper. Please see Page 14, line 22-23.

4.Figure 4. The term ‘PLE’ was not defined. Also, in the Introduction, the acronym ‘NCs’ was used but not defined earlier.

Response:

We have define the term “PLE” in the revised manuscript. Please see Page 4, line 7.

5.Two typos in Figure 7 needed to be corrected: Ea——>Eb and Mev——> MeV. Also, the PL intensity is observed to be increasing with decreasing temperature. However, it slightly drops at T=278 K. Can the authors provide any qualitative explanation?

 Response:

We have corrected two typos in Figure 8. Please see Page 10, Figure 8. It is still unclear why the PL intensity slightly drops at T=278 K. We would continue to conduct it in the future.

6.I suggest replacing the ‘rare earth doped’ with ‘rare-earth-doped’ in the Abstract.

Response:

We sincerely thank the reviewer for his/her valuable suggestion. We have replaced the ‘rare earth doped’ with ‘rare-earth-doped’ in the Abstract..

Reviewer 2 Report

The modern perovskite-like materials shows a wide field for use in modern technology, primarily due to their uniques electron-optical properties. Therefore, the topic discussed in this article “Synthesis and improved photoluminescence of SnF2-derived 3 CsSnCl3-SnF2:Mn2+ perovskites by rapid thermal treatment»  is very relevant for many researchers in the field of synthesis of new compounds. The article submitted for consideration makes a good impression both in the amount of experimental material and in the quality of its presentation. However, I have several questions, primarily related to the application of rapid thermal treatment (RTT) technology:

1. What is the advantage of this technology compared to classic solid state synthesis technology?

2. What determines the exposure time of 60-300 seconds? Why not one hour or one day - perhaps this gave material with even higher parameters, which is demonstrated in the presented work.

3. What do high heating and cooling rates give in terms of the quality of the obtained samples of the Mn2+-doped CsSnCl3?

In my opinion, the presented material could be supplemented with answers to the questions presented. The article makes a very good impression and can be published in the journal "Materials" after these minor revision.

Author Response

  1. What is the advantage of this technology compared to classic solid state synthesis technology?

Response:

We sincerely thank the reviewer for his/her valuable suggestion. Due to the poor thermal stability of tin-based halide perovskite materials, traditional thermal annealing techniques can not effectively activate Mn ion luminescence centers in these materials at high temperature due to the structural decomposition under long time thermal treatment. Rapid heat treatment (RTT) is a transient process characterized by rapid heating and cooling, with short heating time and fast cooling rate. As shown in Figure 6 in the manuscript, the diffraction peaks corresponding to (400), (020), (411), and (002) crystal planes of cubic CsSnCl3 after the RTT process became stronger and sharper with an increase in the RTT time from 60 s to 300 s. This strongly indicates an improved crystallinity of the CsSnCl3:Mn2+ powders after the RTT process. On the other hand, the excitation peaks of CsSnCl3:Mn2+ at 355, 418, and 469 nm, corresponding to the  6A1(6S)→4T2(4D), 4T2(4G), and 4T1(4G) transitions of the Mn2+ ion, respectively, were observed after the RTT process,, as shown in Figure 4 in the manuscript. This strongly indicates that the RTT process effectively activates Mn2+ in CsSnCl3. These results clearly demonstrate that RTT not only can avoid thermal decomposition of tin-based halide perovskite materials during the heating process but also improve the crystallinity of the CsSnCl3:Mn2+ powders, and provides an alternative method for thermal activation of Mn ion luminescence center of tin-based perovskite. We have added the advantage of RTT in the revised manuscript. Please see Page 12, line 3-7.

  1. What determines the exposure time of 60-300 seconds? Why not one hour or one day - perhaps this gave material with even higher parameters, which is demonstrated in the presented work.

Response:

Thank you for your suggestions. In fact, in this work, we have systematically studied the effect of rapid heat treatment time ranging from 30 s to 500 s on the luminescence properties of CsSnCl3-SnF2:Mn2+. It was found that the luminescence properties deteriorated rapidly when the heat treatment time exceeded 300s.

  1. What do high heating and cooling rates give in terms of the quality of the obtained samples of the Mn2+-doped CsSnCl3?

Response:

In our case,the RTT process was carried out on the Rapid Thermal Processor we designed, the samples were heated to the annealing temperature of 200 °C at a rate of 10 °C s-1. We found that when the heating rate is raised to 12 °C s-1, the annealing temperature cannot be maintained steadily at 200 °C, which leads to the failure of the experiment. Therefore, it is not clear how high heating and cooling rates affect the quality of Mn2+ doped CsSnCl3 samples. We will continue this research after improving the rapid heat treatment equipment.

Reviewer 3 Report

In this manuscript, the authors reported that a rapid synthesis method for producing CsSnCl3:Mn2+ perovskites, derived from SnF2, and investigate the effects of rapid thermal treatment on their photoluminescence properties. The authors claimed that the luminescence dynamics of Mn2+-doped CsSnCl3 and open up new possibilities for controlling and optimizing the emission of rare earth doped CsSnCl3. In overall, this manuscript is interesting but in order to consider publication, this work should be revised. The following comments should be addressed for the improvement of their manuscript.

Comment 1: The overall study aims for this novel production method of CsSnCl3:Mn2+ perovskites, derived from SnF2 need to be further clarified in detail as compared to current conventional system for perovskites production.

Comment 2: The various recent reports and their research findings on the “production of CsSnCl3:Mn2+ perovskites, derived from SnF2” and their resultant photoluminescence properties should be summarized into a table form and discussed for better understanding in term of benchmarking points with your research findings.

Comment 3: What is the role of rapid thermal treatment in the rapid synthesis for producing CsSnCl3:Mn2+ perovskites in controlling the photoluminescence properties? Please discuss and clarify with fundamental support. 

Comment 4: The raman analysis is needed to provide detailed information about chemical structure, crystallinity and molecular interactions between for the sample of CsSnCl3:Mn2+ powers with and without RTT process.

Comment 5: The carefully English correction is necessary for the whole manuscript. Please check and revise accordingly.

The carefully English correction is necessary for the whole manuscript. Please check and revise accordingly.

Author Response

1.The overall study aims for this novel production method of CsSnCl3:Mn2+ perovskites, derived from SnF2 need to be further clarified in detail as compared to current conventional system for perovskites production.

Response:

We sincerely thank the reviewer for his/her valuable suggestion.  We have added the corresponding information in the revised manuscript. Please see Page 2, line 19-22.

2.The various recent reports and their research findings on the “production of CsSnCl3:Mn2+ perovskites, derived from SnF2” and their resultant photoluminescence properties should be summarized into a table form and discussed for better understanding in term of benchmarking points with your research findings.

Response:

Thank you for your suggestions. We agree well with you. However, we cannot find any information about the “production of CsSnCl3:Mn2+ perovskites, derived from SnF2” and their resultant photoluminescence properties”. In this paper, we first reported the synthesis and photoluminescence properties of SnF2-derived CsSnCl3:Mn2+

3.What is the role of rapid thermal treatment in the rapid synthesis for producing CsSnCl3:Mn2+ perovskites in controlling the photoluminescence properties? Please discuss and clarify with fundamental support. 

Response:

This question has been described in Reviewer #1 points (1) above.

4.The raman analysis is needed to provide detailed information about chemical structure, crystallinity and molecular interactions between for the sample of CsSnCl3:Mn2+ powers with and without RTT process.

Response:

Thank you for your suggestions. We have added the raman analysis in the revised manuscript. Please see Page 9, line 3-5 and Figure 7.

Figure 7 Raman spectra of CsSnCl3:Mn2+ powers with and without RTT process.

5.The carefully English correction is necessary for the whole manuscript. Please check and revise accordingly.

Response:

Thank you for your suggestions. We have carefully polished the English Language. 

Round 2

Reviewer 3 Report

In overall, this manuscript was technically well revised. This revised manuscript meets the criteria of Materials MDPI. Therefore, in my opinion, the revised manuscript can be accepted for publication.

Author Response

Thank you for the reviewer's comments